# Historical Variation of IEA Energy and CO₂ Emission Projections: Implications for Future Energy Modeling

**Luís M. Fazendeiro** [1,*] and **Sofia G. Simões** [2]

1. CENSE—Center of Environmental and Sustainability Research, NOVA School of Science and Technology, Campus de Caparica, NOVA University Lisbon, 2829-516 Caparica, Portugal
2. LNEG—Laboratório Nacional de Energia e Geologia, I.P. Estrada da Portela, Bairro do Zambujal Ap 7586, 2720-999 Amadora, Portugal; sofia.simoes@lneg.pt
* Correspondence: l.fazendeiro@campus.fct.unl.pt

**Abstract:** The World Energy Outlook reports produced by the International Energy Agency have long been considered the "gold standard" in terms of energy modeling and projecting future trends. It is thus extremely important to assess how well its projections are aligned with sustainable development goals as well as closely tracking observed, historical values. In this work we analyzed thirteen sets of World Energy Outlook projections from the last 25 years. Different scenarios were considered for the following regions and countries: world, OECD, OECD Europe, OECD North America, China, India, Russia, and Africa. The maximum variation between the projections for 2030 CO₂ emissions from the energy sector, made between 2006 and 2018 for OECD, Europe and North America were found to be comparable with the gap between the Paris Agreement goals and the voluntary (unconditional) nationally determined contributions to remain below a 2 °C global temperature increase. For the same period, projections for the percentage of renewable electricity exhibited maximum variations between 51% and 96%, signaling a huge underestimation. We discuss the significance of overestimating energy demand and underestimating the rate of renewable energy implementation in the context of 2030 climate and energy policy targets, as well as desirable methodological changes to energy modeling under aggressive climate mitigation policies.

**Keywords:** energy system modeling; climate change mitigation; global and regional energy projections; nationally determined contributions; integrated energy system planning

## 1. Introduction

Energy systems modeling is now a well-established field of research and an essential activity for modern societies, contributing to the optimization of economic activities and also to minimize the associated environmental impacts [1]. After more than four decades of extensive modeling at international, regional, national, and even local level, energy systems modeling has arguably achieved a very high degree of maturity [2]. However, the large number of projections accumulated during this time-period also raises the interesting question of how well energy projections overall have matched observed historical developments [3]. In this context, several authors seem to concur that energy scenarios are more relevant in terms of the qualitative guidelines they can provide for policy implementation than any quantitative predictions they might contain [4–10].

Probably the most well-known and influential long-term energy projections are the ones developed by the International Energy Agency (IEA). Since 1977 the IEA has been publishing energy demand projections for the global economy as a whole, as well as for several regions and larger individual countries, in the World Energy Outlook (WEO) reports. These projections were originally developed with the goal of supporting energy policy making and ensuring energy security, particularly for the Organization for Economic Co-operation and Development (OECD) group of nations [3]. They have since become the "gold standard" for global and regional energy projections [3,11].

In our view, the main reasons for this are: the fact that the WEO is published annually (the only exception since 1994 was for the year 1997); the scope and credibility of the IEA organization, which is possibly the largest intergovernmental organization in this area of expertise; and, finally, the level of technical detail and regional disaggregation considered in successive WEO reports. These arguments in turn have given the WEO projections an unprecedented level of authority, particularly in the eyes of policy makers [11,12]. Although several other sources for projections on energy demand, energy technologies, and carbon dioxide ($CO_2$) emissions are widely available in the literature, they either: (i) originate from oil and gas companies, with a very large and direct financial interest in the results of their own projections, such as BP, Shell, Statoil, or ExxonMobil, and can thus be reasonably suspected of at least some bias; (ii) offer very little regional disaggregation, lack technical detail, or rely mainly on data gathered by other agencies; or, (iii) have only been published for the first time in recent years, as is the case for recent reports from Greenpeace [13], the World Wide Fund [14], or the World Energy Council [15], and in most cases lack the continuity of regular (yearly) updating.

Therefore, we analyze in this work the WEO projections made by the IEA during the last two and a half decades, focusing on their Current Policies and Reference Scenario. This builds on recent work by Cabeza et al. [3], in which the authors discussed the general trends in total energy demand and energy intensity within OECD countries and China from 1977 to 2013, and how the observed values compared with successive IEA projections. In the present work we go one step further by also including three other regions: India, Russia, and the African continent, since all these will be crucial in determining global energy consumption patterns for the foreseeable future. We also consider a (larger) set of thirteen WEO reports, instead of only the five (1977, 1982, 1994, 1998, 2004) considered by Cabeza et al. We further disaggregate the OECD into OECD Europe and OECD North America and quantify the variation in the projections of total primary energy demand, the associated $CO_2$ emissions from the energy sector, and the share of electricity from renewable energy sources for these eight regions (world, OECD, OECD Europe, OECD North America, China, India, Russia, and Africa).

In another article by Carrington and Stephenson, the authors performed a critical assessment of the projections from the IEA and other agencies, with an exclusive focus on the rates of solar PV deployment [11]. They found that "the influential IEA scenarios [which] have conservative projections for solar PV growth and offer little analysis of emerging energy technologies", concluding that the WEO projections for solar PV consistently underestimated the actual implementation rates. In this article we have chosen to focus instead on the percentage of electricity generated from renewable technologies as a proxy indicator for the transition to low carbon energy systems, keeping in mind the crucial role that renewable electricity is widely expected to have in deep decarbonization scenarios [16–21].

Similar conclusions were reached by Metayer et al. who noted that the IEA assumed a linear growth trend for solar and wind technologies, whereas historical data clearly favored a long-term exponential growth rate for new renewable technologies [22]. These authors performed the most comprehensive analysis of WEO projections that we were able to find in the literature, by processing data from 15 different reports published between 1994 and 2014, and examining trends for total primary energy demand (TPED) for renewable technologies, such as solar PV, electricity demand, and nuclear energy. However, this 2015 work did not include any projections published after the Paris Agreement was approved. We believe that it is extremely important to update this assessment in view of more recent WEOs and the urgent need for more stringent emission reduction goals, and to discuss how the variation in the projections compares with the targets for greenhouse gas (GHG) emissions in 2030, as we do in this work.

In this context, several reports from NGOs and public advocacy groups have in recent years been quite critical of the IEA projections, particularly in relation to its likely overestimation of fossil fuel demand and underestimation of the observed rate of renewable

energy implementation. We note in particular the excellent work of the Carbon Tracker Initiative which played a major role in establishing and widely publicizing the concept of the "carbon bubble" (or "unburnable" fossil fuel assets) starting from their first 2011 report [23]. One particular milestone was their 2017 report, which convincingly argued that the potential for energy demand reduction might be systematically underestimated by the baseline scenarios from the IEA, BP, ExxonMobil, and other international agencies and large companies [24]. In the present work we go one step further by directly comparing the 2030 maximum variations for energy sector $CO_2$ emissions found in the WEO reports, with the emissions gap between the projections and the climate mitigation targets compatible with a global warming of less than 2 °C.

Since the Paris Agreement (PA) was adopted in December 2015 [25], several authors have analyzed the nationally determined contributions (NDCs) submitted by most of the 196 countries which are part of the Agreement [26–29], with particular reference to the United Nations Environment Programme (UNEP) Emissions Gap Report [30]. All these different analyses concur that the current NDCs are not enough to comply with the PA targets, with further ambition urgently required. Thus, in this work we compare the WEO projections for $CO_2$ emissions from the energy sector with the emission targets for 2030 for six of the eight regions defined (world and Africa were not considered, for methodological reasons). We also focus on a precise time interval of 12 years (2006–2018), equal to the time interval separating 2018 (the date of the last WEO report analyzed in this work) from 2030. Our main research question is to assess whether the maximum variation in the different projections is comparable with the (relative) gap between the emissions expected in 2030 and the reduction levels needed to avoid 2 °C average global heating, compared to pre-industrial levels. To the best of our knowledge, it is the first time that such an analysis has been carried out.

## 2. Materials and Methods

### 2.1. Data and Regions Considered

Our analysis focused on energy projections made for the years 2010, 2020, 2030, and 2040, based on data published by the IEA in its WEO reports [31–43]. We chose to only include projections starting from 1994, since before that date there were only two WEO reports, one in 1977 and another in 1982. The reports analyzed, as well as the years they cover, are listed in Table 1. The regions and countries considered in this analysis are the world, OECD, OECD Europe, OECD North America, China, India, Russia, and Africa (see Table A1). For each of these regions we quantified the variations in thirteen editions of the WEO reports (Table 1) focusing on the key indicators: TPED, $CO_2$ emissions from the energy sector, and the percentage of RES electricity.

For the WEO reports covering the period between 1994 and 2008, we considered the IEA "Reference" Scenario (also called "Capacity Constraints" in the 1996 edition [32] and "Business as Usual" (BAU) in 1998 [33]). Starting from 2010 this was replaced with a "Current Policies" Scenario (Table 1). Two extra scenarios also introduced in 2010, but not considered in the present work, are "New Policies" and "450". Instead, we deliberately chose to focus on the more conservative scenarios in each WEO edition (Reference/Current Policies). The reason for this choice is that these scenarios and their respective projections are widely regarded as the "gold standard of energy analysis" and are extremely influential in terms of public perception and policy making [3,11,12,22,23]. We note that the Current Policies Scenario does not yet include the commitments already made by countries in the form of the NDCs. What it does assess is, taking the WEO 2018 as an example, "the impact of those policies and measures that are firmly enshrined in legislation as of mid-2018" [38]. Details about the "World Energy Model" which generates these projections can be found in the IEA documentation [44]. The exact definition for the 8 regional groupings can be found in Appendix A. Moreover, for a general economic overview of the considered regions and several key energy indicators, see Appendix B.

**Table 1.** List of IEA WEO reports analyzed in this paper.

| Publication Year | Years for Which Data or Projections Are Given | IEA Scenario |
|---|---|---|
| 1994 [31] | 1971 *, 1991 *, 2000, 2010 | "Reference" Scenario |
| 1996 [32] | 1971 *, 1993 *, 2000, 2010 | |
| 1998 [33] | 1971 *, 1990 *, 1995 *, 2010, 2020 | |
| 2000 [34] | 1971 *, 1997 *, 2010, 2020 | |
| 2002 [35] | 1971 *, 2000 *, 2010, 2020, 2030 | |
| 2004 [36] | 1971 *, 2002 *, 2010, 2020, 2030 | |
| 2006 [37] | 1990 *, 2004 *, 2015, 2030 | |
| 2008 [38] | 1990 *, 2006 *, 2015, 2020, 2025, 2030 | |
| 2010 [39] | 2020, 2030, 2035 | "Current Policies" Scenario |
| 2012 [40] | 2020, 2030, 2035 | |
| 2015 [41] | 2020, 2030, 2040 | |
| 2017 [42] | 2025, 2030, 2040 | |
| 2018 [43] | 2025 2030, 2040 | |

* Historical data, per time of publication.

### 2.2. Key Indicators and Methods

In this work, we developed an approach to assess the variation within the IEA WEO projections, by considering a two-step approach. In Section 3.1 we look at some of the projections made for the year 2010 from six different WEO reports in order to assess how close they were to what was actually observed, and in which areas these projections were further off. In Section 3.2 we extend this to also cover projections made for future years, with an emphasis on 2030 and 2040. Finally, in Section 3.3 we zoom in on the year 2030 and compare the projections made by different WEOs for $CO_2$ emissions from the energy sector with the current NDCs for that year. We thus linked the projections made for past years with the projections for the future and tried to translate those findings into possible implications for the uncertainty of future projections for 2030 (in particular) and 2040.

The three key indicators used in this exercise were TPED, $CO_2$ emissions from the energy sector, and the total consumption of electricity (in Section 3.1) or percentage of electricity generated from renewable sources (in Section 3.2). We believe that these indicators can give us an overall picture of how energy systems are evolving. They have also been utilized in recent similar studies. As an example, the authors of [3] looked at global and regional total primary energy supply and total final energy consumption, comparing per capita and per GDP projections (also from several WEO reports) with historical values. As for the role of electricity, it is now widely acknowledged that it must play a crucial role in any decarbonization strategy [16,18,21,45,46]. If, until recently, the main focus was on electrification, regardless of the source of electricity, in order to increase access to energy [17], it is becoming increasingly clear that, in order to fulfil the Paris Agreement goals, all newly installed electricity (and, progressively, also the substitution of existing fossil fuel power plants) will have to come from renewable sources [16,45,46].

### 3. Results

#### 3.1. Variation in Projections between 1994 and 2010

In this section we compare some of the assumptions underlying energy projections made in the WEO reports for the years between 1994 and 2010 [31–36], such as population and GDP growth, with the historical variations de facto observed. The choice of 1994 and 2010 for the time interval stemmed from the fact that the 1994 WEO is the first WEO considered in this work (Table 1), and 2010 is the one for which more projections are available (a total of six) from the list of WEO reports considered (Table 1).

Table 2 shows a comparison for the period 1994–2010 between the historical rate of growth (left columns) in both population and GDP per purchasing power parity (PPP) in 2010 and in USD billion, and the average (right columns) of all the variations projected by those six WEO reports. As an example, the WEO average projections for world population

growth in 2010 were aligned with the historical value (23%), whereas for GDP the WEO projections tended to slightly underestimate economic growth (actual value was the same as the maximum found in all projections). In the case of Africa, China, and India, even the maximum WEO projections for economic growth were all below what was actually observed, something which we discuss in the next section. We note however that no comparable mismatch occurred for population growth projections, with historical values falling next to the average for projections for these same three regions.

**Table 2.** Historical population (blue) and GDP (red) growth between 1994 and 2010, compared with the average of WEO projections made for 2010 between 1994 and 2004.

| Region | Historical Variation '94–'10 | | Average of WEO Projections for 2010 | |
|---|---|---|---|---|
| | Population | GDP PPP (USD Billion in 2010) | Population | GDP PPP (USD Billion in 2010) |
| World | 23% | 79% | 23% | 65% |
| OECD | 12% | 43% | 8% | 44% |
| OECD Europe | 7% | 40% | 3% | 42% |
| OECD N. America | 20% | 49% | 14% | 46% |
| China | 12% | 358% | 15% | 169% |
| India | 31% | 198% | 28% | 118% |
| Russia | −4% | 66% | −7% | 73% |
| Africa | 49% | 106% | 46% | 74% |

Table 3 shows the results of a similar analysis, but this time for the variation observed (left) and projected (right) by the same six WEO editions for TPED, $CO_2$ emissions from the energy sector, and the total consumption of electricity. Overall, we see a fairly good agreement between the projections for TPED, $CO_2$ emissions, and the actual historical variations, although China is again an outlier. However, a clear trend can be seen in terms of the projections for total electricity. In all regions except for China, the historical variation stayed below the minimum values (not shown here) projected in the six WEO reports. In Section 3.2.3 we will come back to this fact when discussing the share of renewable electricity projected by the Reference Scenario of the WEOs. In the case of China, the variation in the total consumption of electricity (354%) was much larger than even the maximum WEO projections.

**Table 3.** Historical growth of TPED (blue), $CO_2$ emissions from energy (pink), and the total electricity consumption (green) between 1994 and 2010 compared with the average of WEO projections made for 2010.

| Region | Historical Variation '94–'10 | | | Average of WEO Projections for 2010 | | |
|---|---|---|---|---|---|---|
| | TPED | CO₂ Emissions (from Energy) | Total ELC | TPED | CO₂ Emissions (from Energy) | Total ELC |
| World | 43% | 47% | 68% | 35% | 44% | 73% |
| OECD | 14% | 9% | 32% | 17% | 19% | 41% |
| OECD Europe | 12% | 0% | 31% | 17% | 16% | 42% |
| OECD N. America | 11% | 10% | 27% | 19% | 23% | 41% |
| China | 161% | 200% | 354% | 62% | 84% | 172% |
| India | 99% | 142% | 145% | 91% | 145% | 209% |
| Russia | 5% | −4% | 19% | 7% | 8% | 35% |
| Africa | 64% | 80% | 87% | 35% | 96% | 114% |

In order to analyse such trends, we then measured the variations in these same three indicators (from Table 3), in per capita (Table 4) and per GDP (Table 5) terms. These two sets of data thus combine the data shown in Tables 2 and 3, regarding what the six different WEO reports were projecting in their Reference Scenario for five variables: population, GDP, TPED, $CO_2$ emissions, and total consumption of electricity.

**Table 4.** Per capita variations of historical TPED (blue), CO$_2$ emissions from energy (pink), and total electricity consumption (green) for 1994–2010 (left) compared with the average of WEO projections for the year 2010 (right).

| Region | Historical Variation '94–'10 (per Capita) | | | Average of WEO Projections for 2010 (per Capita) | | |
| --- | --- | --- | --- | --- | --- | --- |
| | TPED | CO$_2$ Emissions (from Energy) | Total ELC | TPED | CO$_2$ Emissions (from Energy) | Total ELC |
| | | | | Avg | Avg | Avg |
| World | 16% | 19% | 36% | 10% | 17% | 41% |
| OECD | 1% | −3% | 17% | 10% | 11% | 31% |
| OECD Europe | 5% | −6% | 23% | 14% | 12% | 38% |
| OECD N. America | −7% | −8% | 6% | 4% | 8% | 23% |
| China | 132% | 167% | 305% | 41% | 60% | 138% |
| India | 53% | 85% | 87% | 50% | 90% | 140% |
| Russia | 9% | −1% | 24% | 16% | 17% | 45% |
| Africa | 10% | 21% | 25% | −7% | 34% | 47% |

**Table 5.** Economic intensity (per USD thousand in 2010) of TPED (blue), CO$_2$ emissions from energy (pink, and total electricity consumption (green) for 1994–2010 (left), compared with the average of WEO projections for the year 2010 (right).

| Region | Historical Variation '94–'10 (per USD 1000 in 2010) | | | Average of WEO Projections for 2010 (per USD 1000 in 2010) | | |
| --- | --- | --- | --- | --- | --- | --- |
| | TPED | CO$_2$ Emissions (from Energy) | Total ELC | TPED/1000 USD 2010 | CO$_2$ Emissions (from Energy)/ USD 1000 in 2010 | Total ELC/ USD 1000 in 2010 |
| World | −20% | −18% | −6% | −18% | −12% | −10% |
| OECD | −21% | −24% | −8% | −18% | −17% | −16% |
| OECD Europe | −20% | −28% | −7% | −17% | −18% | −14% |
| OECD N. America | −26% | −26% | −15% | −19% | −16% | −17% |
| China | −43% | −35% | −1% | −37% | −30% | −9% |
| India | −33% | −19% | −18% | −11% | 13% | 22% |
| Russia | −37% | −42% | −29% | −37% | −36% | −32% |
| Africa | −20% | −13% | −10% | −21% | 13% | 6% |

From Table 4 we see that almost the same exact trends observed in Table 3 are on display. In particular, the projections for the variation in the total electricity consumption per capita (green, right) are still above the historical variation in all regions, except for the case of China. Interestingly, per capita CO$_2$ emissions from the energy sector fell in the three OECD regions whereas the WEO reports had projected they would grow, a trend also seen in Table 3.

When we turn to variations for the same key indicators, but in terms of economic intensity, shown in Table 5, all the averages of the WEO projections for TPED (blue, right) are now very close to the historical variation, except for the case of India. The same is true for CO$_2$ emissions (pink). Once again, the case of China is of particular interest. Although its actual growth in TPED had been larger than all projections (either in absolute terms, Table 3, or per capita, Table 4), we now see that the historical variation (−35%) is very close to the average of the projections (−30%). Even though GDP growth in the 1994–2010 period widely surpassed all WEO projections (Table 2), as did growth in the three key indicators considered (Table 3), dividing these values cancels the variations. This could not happen in the case of per capita indicators (Table 4) because the projections for population growth in China were very close to the historical variation (Table 2).

### 3.2. Variation in Projections from 1990 to 2040

We now turn to the variations observed in the IEA projections up to 2040. In this section we present the maximum variations found in over two and a half decades of WEO projections of TPED, CO$_2$ emissions from the energy sector, and the percentage of RES

electricity for the regions considered in this work (see Tables 1 and A1). This analysis is then followed by a comparison with the variations encountered and the 2030 NDCs, in order to assess if the variations in energy projections and their associated emissions might be comparable to the gap between the current NDCs and the Paris Agreement goals. As discussed in previous sections, we should keep in mind that the regions considered had very different economic and social trajectories in the past decades.

### 3.2.1. TPED

In Figure 1 we show the values for TPED in GWh for the regions chosen for this analysis for the IEA "Reference" [31–38] and "Current Policies" [39–43] scenarios, for the years between 1990 and 2040. Some general features can be readily identified related to major historical events, including: the fall of the Soviet Union, which greatly influenced Russia's TPED, leading to a decrease of about 35% between 1990 and 1997; the financial crisis of 2008–2009, which more clearly affected the OECD (and in particular Europe) than the rest of the world (TPED fell 3.3% in OECD Europe between 2008 and 2013, compared to a decrease of only 1.8% for the whole of OECD and an increase in Chinese TPED of 42.5% for the same period); or the spectacular rise in China's TPED by 335% between 1997 and 2013, which seemed to have caught modelers by surprise, as so often is the case with events of such magnitude [4].

In this respect, it is interesting to observe (Figure 1e) that in 2004 the IEA was predicting a value of TPED for China in 2030 which was reached as early as 2010 (whilst the value of TPED projected in 2004 for China in 2010 ended up being roughly half of the actual value). There was also an important change in the accounting methods of the WEO in 2000, which began to also include "Combustible Renewables and Waste" in TPED, which explains the large shift in values seen in the case of India and Africa (Figure 1f,h).

In Figure 2 we quantified the maximum relative variations (in percentage) found in these different TPED projections, made for the years 2010, 2020, 2030, and 2040, in reports from three different (and increasingly shorter) time intervals: 1994–2018, 2006–2018, and 2015–2018, including a total of 13, 7, and 3 WEO sets of projections, respectively (Table 1). As expected, the magnitude of the variations decreases with time (since different projections made closer in time and using similar assumptions will usually—minus some major geopolitical or economic turmoil—be more in accordance). But we note that there can be quite significant differences in these projections, as pointed out in previous research(e.g., the authors in [4,6,47]). As an example, the projections for TPED in the OECD for the year 2030 went from 83 GWh in 2002 to 63 GWh in 2017 and 2018, a difference of more than 24%. For China, India, and Africa we see even larger variations over a similar time period (although in an opposite direction). This is partially because of much higher growth rates registered in these regions compared to more developed economies (as seen in Table A2, last column), although some issues with the accuracy of data gathering may also be involved.

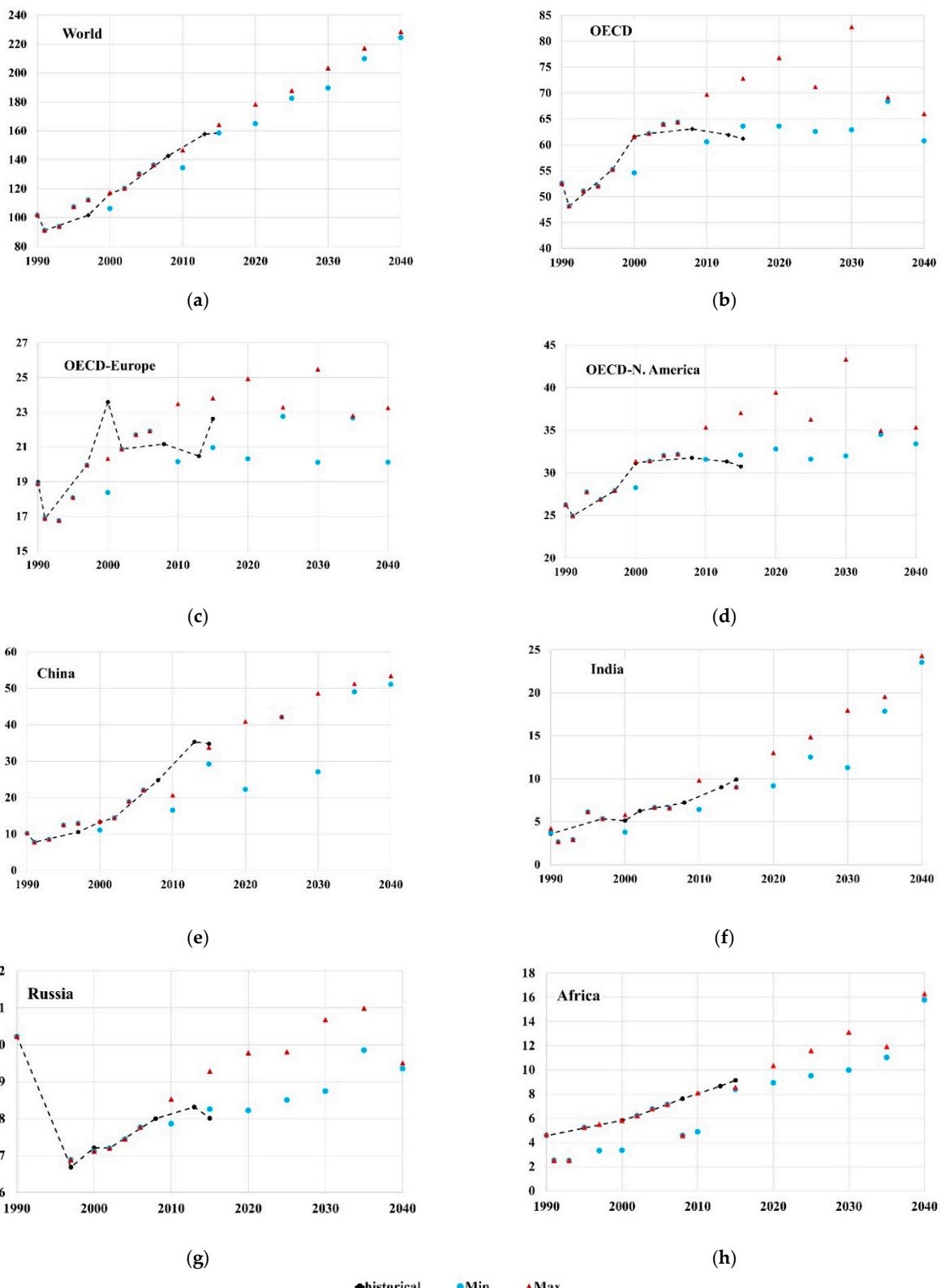

**Figure 1.** Historical values (black dotted line) and WEO projections (minimum and maximum values) for yearly Total Primary Energy Demand (TPED), in millions of GWh, from 1990 to 2040, for 8 regions, from top to bottom and left to right: (**a**) world, (**b**) OECD, (**c**) OECD Europe, (**d**) OECD North America, (**e**) China, (**f**) India, (**g**) Russia, and (**h**) Africa. Note the different vertical scales. Utilized WEO reports are given in Table 1.

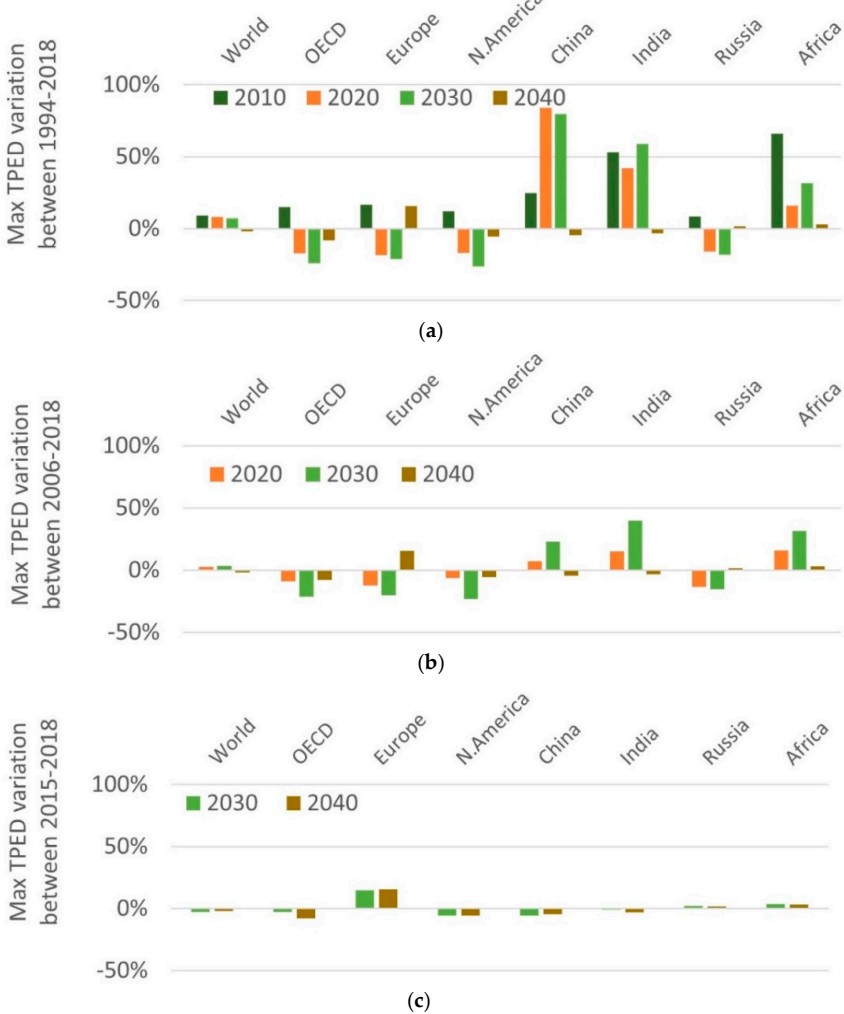

**Figure 2.** Maximum relative variation (in percentage) among the TPED projections for the time intervals (from top to bottom): (**a**) 1994–2018, (**b**) 2006–2018, (**c**) 2015–2018, with respect to the years 2010, 2020, 2030, and 2040. A positive value indicates a correction made upwards, negative values mark downward variations.

Overall, the global projections (world) present smaller variations amongst the different WEO editions (ranging from only −2% for 2040 to 9% for 2010 projections), as expected from Figure 1, but these can mask much larger differences in the projections for individual countries or regions. Moreover, the year for which more projections are considered (nine) is 2030 (see Table 1). The large (relative to other regions) increase in variation within the projections made in 2015–2018 for Europe (Figure 2c) is partly the consequence of the large change that took place in the WEO in 2017 for that regional definition (Table A1). This spike is also observed in Figure 4c, for the only reason that many more countries were included in that regional definition, starting from 2017.

3.2.2. CO$_2$ Emissions from the Energy Sector

Figure 3 shows an analysis similar to the one shown in Figure 1, but this time for CO$_2$ emissions (given in millions of tonnes) from the energy sector, by far the largest source of GHG emissions globally [18]. The projections are given for years between 1990 and 2040. There is a strong correlation with the results shown in both Figures 1 and 3, but with slightly larger (maximum) variations amongst the projections for the latter (Figure 4). As an example, if we focus on the year 2030, the maximum variations found for TPED projections for the eight regions, between 1994 and 2018 (Figure 2a) are, respectively: 7%, −24%, −21%, −26%, 80%, 59%, −18%, and 32%. For CO$_2$ emissions (Figure 4a) they are −8%, −35%,

−32%, −38%, 78%, 88%, −33%, and −38%, so of slightly larger absolute magnitude in all cases, except for China. As was the case for TPED, China and India again present greater variations overall in the projections, particularly for 2030. The large relative variation observed for the region of Europe in recent projections for 2030 and 2040 (Figure 4c) again merely reflects the fact that the number of countries included in that definition greatly increased in the 2017 WEO report.

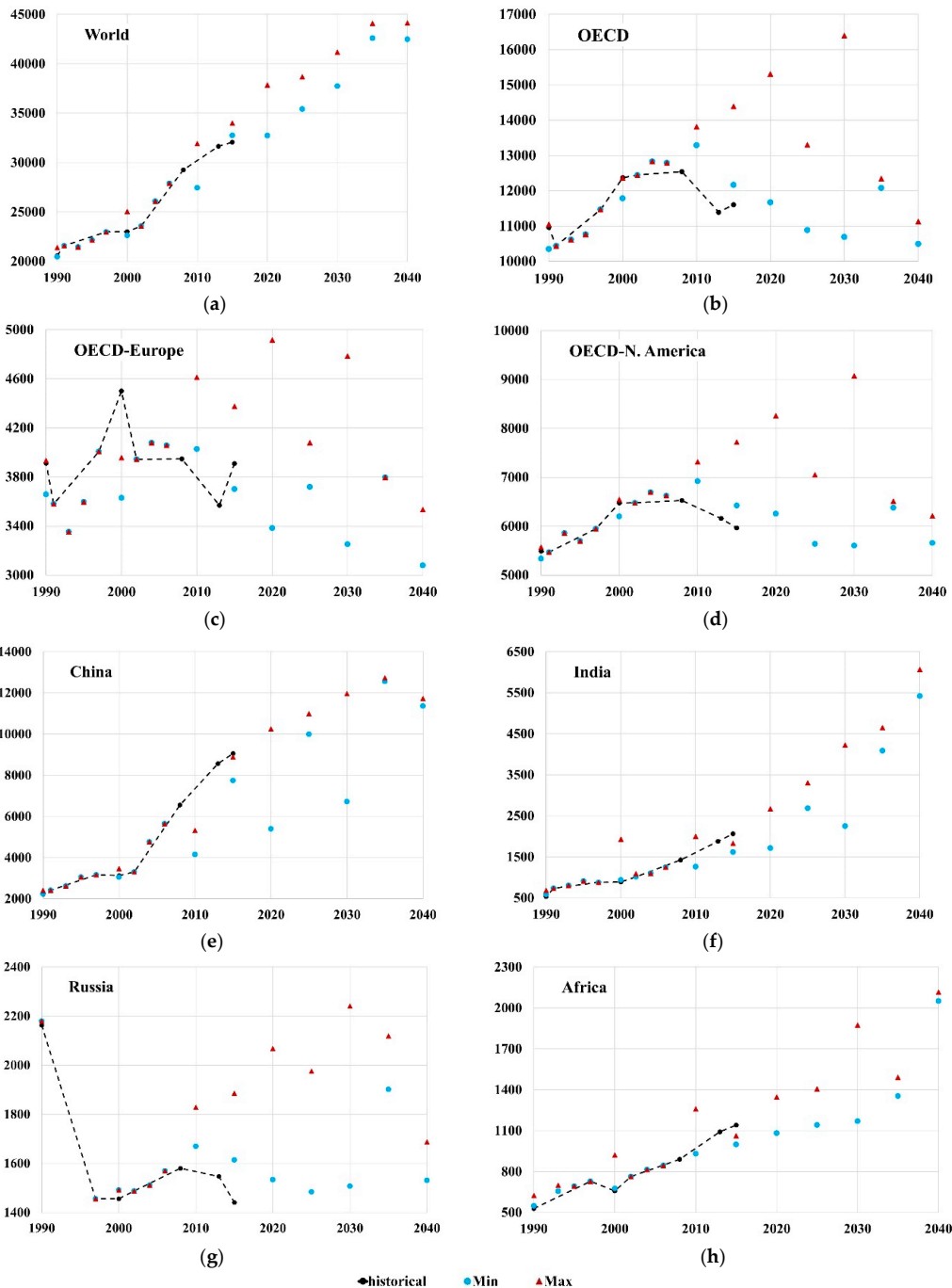

**Figure 3.** Historical values (black dotted line) and WEO projections (minimum and maximum) for yearly emissions of $CO_2$ from the energy sector, in millions of metric tonnes for 8 regions (same as in Figure 1), between 1990 and 2040. Same 8 regions as before, from top to bottom and left to right: (**a**) world, (**b**) OECD, (**c**) OECD Europe, (**d**) OECD North America, (**e**) China, (**f**) India, (**g**) Russia, and (**h**) Africa. Again, note the different vertical scales on each plot.

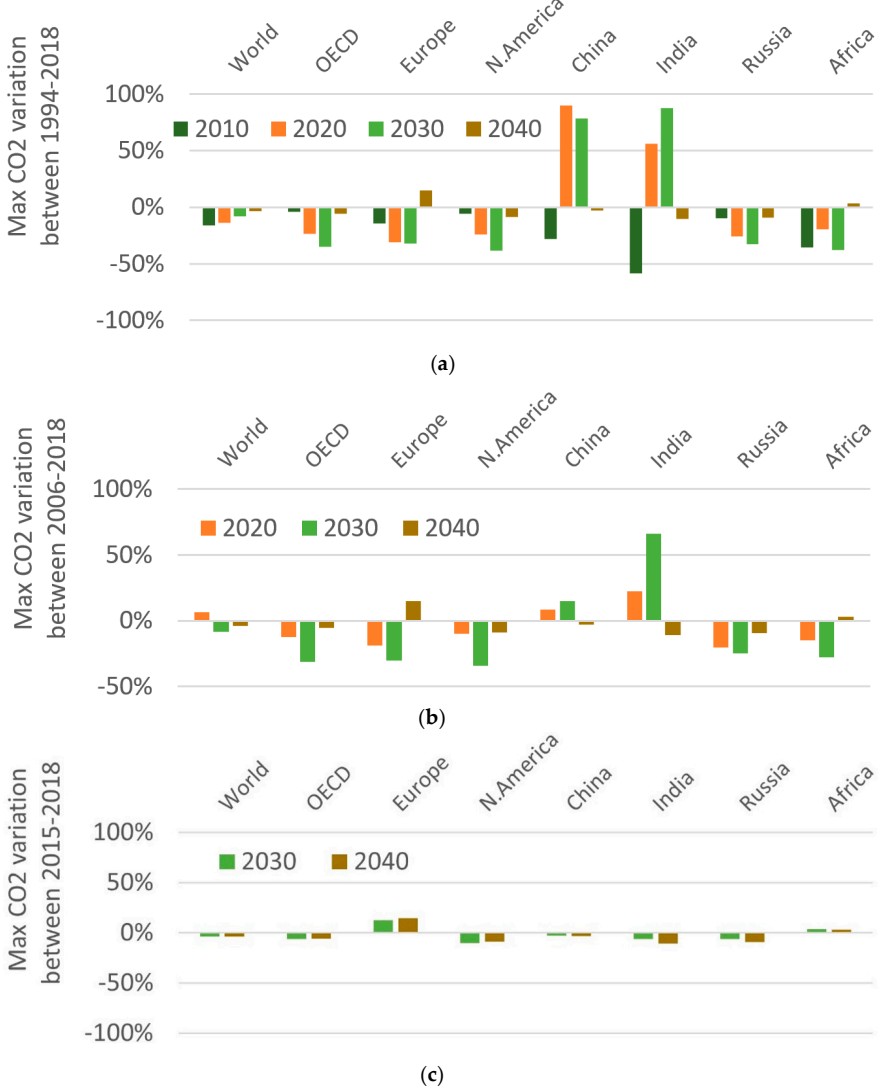

**Figure 4.** Maximum relative variation (in percentage) among the CO₂ projections shown in Figure 3 for the time intervals (from top to bottom): (**a**) 1994–2018, (**b**) 2006–2018, (**c**) 2015–2018, with respect to the years 2010, 2020, 2030, and 2040. The same conventions apply as in Figure 2.

Regarding the differences found between the projections for China and India (particularly for Figures 2a and 4a) and the other regions considered in this work, Table A2 already provides some indication as to why that might be. Namely, between 1990 and 2013, whereas Africa only grew slightly above the world average (138% vs. 115%) in terms of GDP (measured in PPP), China and India grew by a staggering 836% and 325%, respectively. It is to be expected that such large rates of growth would inevitably introduce large variations in in-between projections. It seems obvious that apart from highly disruptive technological changes, the variations found between different TPEDand CO₂ projections can be partly explained by the rate of economic growth observed during the same period. However, we can see from Table 2 that the GDP growth for these two countries was highly underestimated, thus influencing the accurateness of all other related economic projections.

### 3.2.3. Generated RES Electricity

Finally, in Figures 5 and 6 we present a similar analysis, but this time for the share of the electricity generated from RES between 1971 and 2040. We chose to look at generated electricity (i.e., TWh) instead of installed capacity, since the latter may not reflect issues related to intermittency and distribution (i.e., installed sources which may not yet be

connected to the electrical grid). The definition of "renewable" used in this instance, following the IEA WEO convention, considers all sources of electricity not generated using either coal, gas, oil or nuclear, and thus also including hydro.

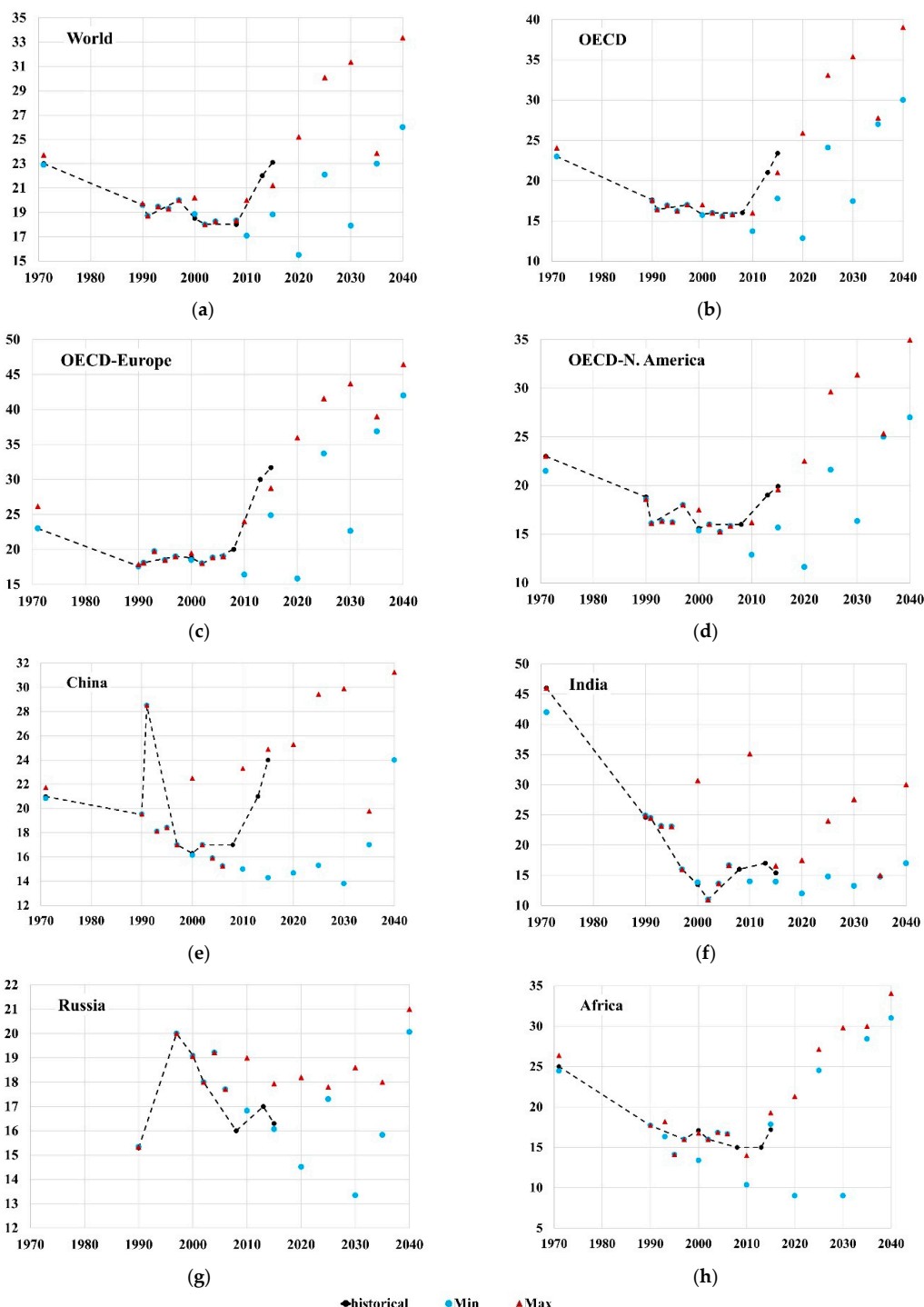

**Figure 5.** Historical values (black dotted line) and WEO projections (minimum and maximum) for the percentage of generated electricity from RES (hydro, geothermal, biomass, solar, wind, etc.) for 8 regions (the same as in Figure 1), between 1971 and 2040. Same 8 regions as before, from top to bottom and left to right: (**a**) world, (**b**) OECD, (**c**) OECD Europe, (**d**) OECD North America, (**e**) China, (**f**) India, (**g**) Russia, and (**h**) Africa. Again, note the different vertical scales.

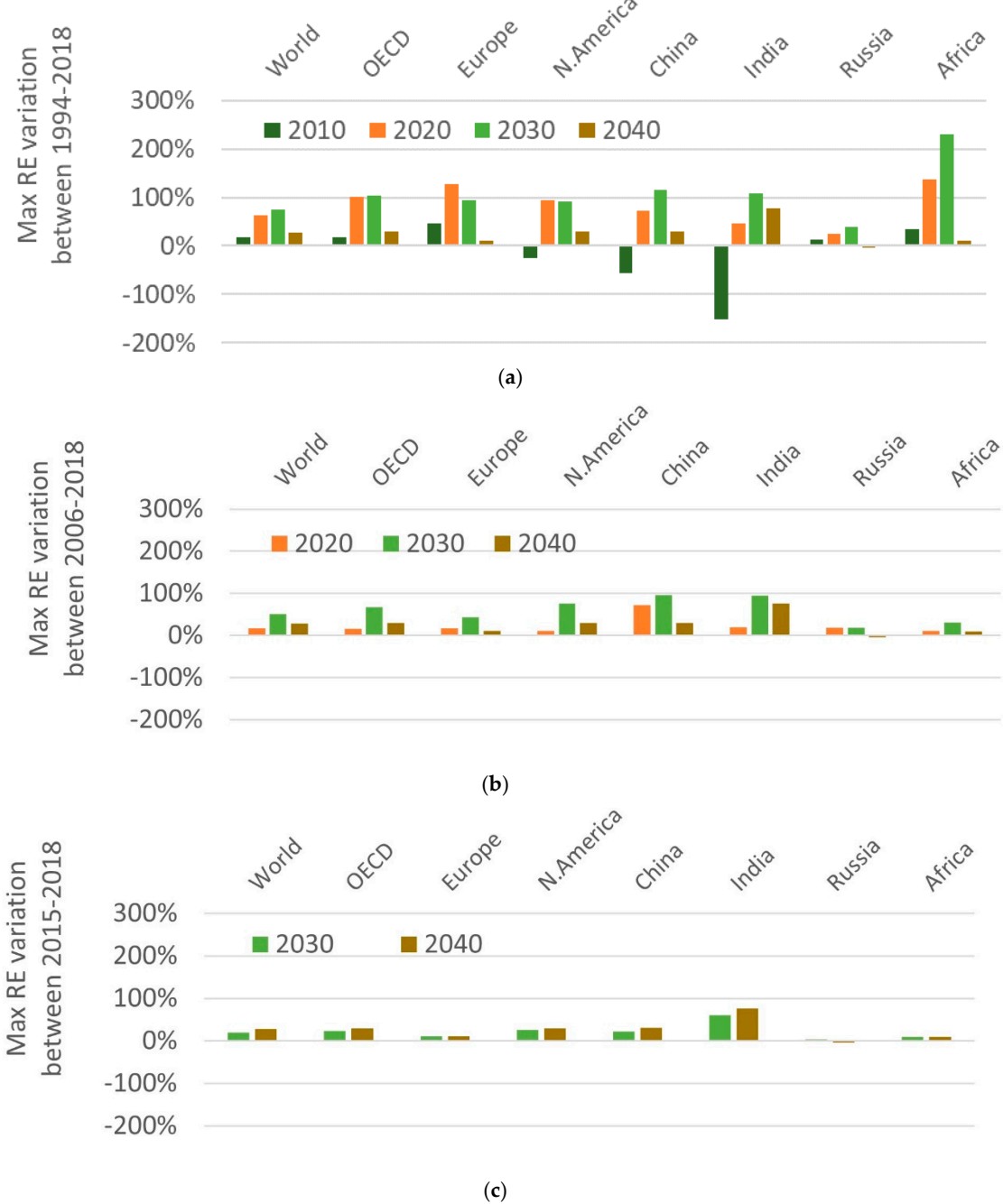

**Figure 6.** Maximum relative variation (in percentage) among the fraction of generated RES electricity, shown in Figure 5, for the time intervals (from top to bottom): (**a**) 1994–2018, (**b**) 2006–2018, (**c**) 2015–2018, with respect to the years 2010, 2020, 2030, and 2040. The same conventions apply as in Figures 2 and 4.

One interesting aspect shown in Figure 5 is the fact that for many regions of the world, such as North America, India, and Africa, the share of renewable electricity was greater in 1971 than in 2015 (and that is why we choose to show the 1971 values in this case, and not in Figures 1 and 3). This can be easily explained by the very large global share of hydroelectricity in energy systems, which has decreased in recent years. As an example, in 1971 46% of the electricity generated in India was from hydro, from a total of 61 TWh [38]). In 2013 this value had increased by a factor of almost 20 to 1193 TWh, but the share of hydro was now only 12%, with other renewable sources (bioenergy, wind, and solar PV) adding another 5% (equivalent to 60 TWh) [41]. During that period, other non-RES sources greatly increased their share in electricity generation in India, with the share of coal rising from

44% in 1971 to 73% in 2013 [36]. A similar evolution occurred in Africa and China. The case of OECD Europe is slightly different, particularly due to the very large increase in RES electricity (particularly wind and solar PV) in the last two decades. In this case, the share of non-hydro RES sources in electricity generation increased from less than 1% in 1971 to 16% in 2015, with another 16% coming from hydro [36,42]. However, this dependency on hydropower is also a dire warning of how much energy security in modern societies is dependent on the level of precipitation, particularly in the context of an increasingly harsh climate change impact on river flows [48].

Another remarkable feature seen in Figure 5 is that the percentage of renewable electricity projected for 2020 in the 2000 WEO was already surpassed, in all the eight regions considered, a few years before 2010. (Even though some regions, such as Russia and India have seen a slight decrease of this percentage in recent years, due to the growth of natural gas and coal, respectively, in electricity generation.) This has been remarked on by several authors and highlights the fact that the IEA reference scenarios have been profoundly conservative and have systematically failed to capture the astonishing pace of renewable energy implementation in recent years [11,49]. It is also interesting to compare these results with Table 3, where we see that the amount of total electricity consumption for all eight regions, with the exception of China, between 1994 and 2010 was lower than the minimum range of WEO projections. Crossing that information with the data shown in Figures 5 and 6 we thus conclude that there was a very large projection, by the WEO reports during that period, of the growth in electricity consumption from non-renewable sources (fossil fuels and nuclear power) that never materialized.

Also, as is clearly seen from Figure 6, the level of maximum variation in the projections for RES electricity is now much greater than it was for $CO_2$ emissions from the energy sector (Figure 4) or for TPED projections (Figure 2), as evidenced by the change in the vertical scale. This also indicates that there appears to be much less variation in the expected evolution of populations and the associated energy demand patterns than on the type of technologies that will supply that demand, as some of the analysis in Section 3.1 alreadyshowed. Besides the magnitude of the variations, these are all (for 2020, 2030, and 2040) now in a positive direction (with only a small exception for a −4% cut in projections for Russia, introduced by the 2018 WEO), indicating successive upwards corrections in successive WEO editions for all eight regions considered. To clarify this point, we can see that the maximum variation in global (world) projections for 2030 is 75%, only falling to 20% in recent years (when comparing Figure 6a with Figure 6c).

### 3.3. Implications for the NDCs

Since the 1970s, energy projections have become increasingly more relevant, not only for energy policy goals (particularly energy security and economic growth), but also for supporting climate policy and the much-needed energy transition to a low carbon system. According to the latest IPCC assessment: "In 2010, the energy supply sector was responsible for approximately 35% of total anthropogenic GHG emissions" [18], by far the largest contributor to GHG emissions. In a prominent article it is argued that the world has roughly three years (starting in 2017) in order to achieve an absolute GHG emissions peak, otherwise the cost of achieving the 2 °C goal becomes increasingly (and prohibitively) steep [45]. In another influential paper, the authors propose that carbon dioxide anthropogenic emissions be halved every decade, once again with an absolute peak being achieved no later than 2020 [46].

The most recent UNEP Emissions Gap Report found a global gap of 12 $GtCO_2e$ (conditional) and 15 $GtCO_2e$ (unconditional) between the current NDCs and the target GHG emissions level in 2030. This is for an emissions' level still compatible with a greater than 66% chance of avoiding a 2 °C warming by 2100 [30]. This would in turn imply reducing total GHG emissions from an expected (under conditional NDCs) level of 56 $GtCO_2e$ to just 41 $GtCO_2e$ (median estimate, in a range of 39–46 $GtCO_2e$), i.e., a global reduction of 27% [30].

If we now compare this variation with the WEO projections for $CO_2$ emissions from the energy sector in 2030, we see that it is either smaller or of a similar magnitude than the maximum variation found for the 2030 projections reported in Figure 4a,b for all regions except "world" (in both Figures) and China (in Figure 4b, which shows a variation of 15%). So, for the regions OECD, OECD Europe, and OECD North America there are variations in the projected $CO_2$ emissions for 2030 of $-35\%$, $-32\%$, and $-38\%$, respectively, for the WEO reports considered between 1994 and 2018 (Figure 4a). The variations are all in the same direction, i.e., in decreasing the projected value in subsequent years. We should again point out that in the WEO Current Policies Scenario, NDCs commitments are still not accounted for. This can in turn be compared with some of the NDCs, e.g., the EU-28 goal of reducing GHG emissions in 2030 by 40% from 1990 levels (recently increased from a 50% target to a 55% target), the commitment by the USA of reducing GHG emissions by 26–28% in 2025 compared to 2005 levels, or Canada's commitment of a 30% reduction in GHG emission by 2030 compared to 2005 [21,23–25]. Globally, the IEA and the WEO variations of world $CO_2$ projections have a much lower magnitude ($-8\%$ for 2030, Figure 4a,b), but this can hide very large regional variations in past projections, as shown in the previous section.

In order to furthercompare these variations, we analyzed several NDCs directly and computed the GHG emissions that they entail for 2030. Table 6 shows the countries whose NDCs were assessed, a very short summary of their goals, and the methodology used to estimate their expected 2030 level of GHG emissions. (In the case of non-ratification of the PA, as per the case of Turkey, these are instead called Intended NDCs. Russia ratified the Agreement in October 2019). For this exercise, no LULUCF contributions were considered. For the case of the USA, China, and India we used the values calculated by the authors in [50] for 2030, assuming an SSP2 "middle of the road" scenario [51]. This stemmed from the fact that the USA NDCs only refer to 2025 and not 2030 (for that country, the authors in [50] assumed a linear extrapolation of emissions reduction rates for the period 2025–2030), and that both China and India refer to the carbon intensity of GDP.

**Table 6.** List of countries and regions whose NDCs were considered with a short summary of the NDCs relating to GHG direct emissions only (LULUCF not accounted for), and the methodology used for estimating GHG emissions in 2030.

| Region or Country | Summary of NDCs | Methodology | Mt $CO_2$e in 2030 |
|---|---|---|---|
| EU-28 | 40% GHG reduction in 2030, compared to 1990 | 1990 emissions, UNFCCC database | 3432 |
| Iceland | 40% GHG reduction in 2030, compared to 1990 | 1990 emissions, UNFCCC database | 2.101 |
| Norway | 40% GHG reduction in 2030, compared to 1990 | 1990 emissions, UNFCCC database | 31.05 |
| Switzerland | 50% GHG reduction in 2030, compared to 1990 | 1990 emissions, UNFCCC database | 22.3 |
| Turkey * | Reduction of up to 21% of GHG emissions from BAU by 2030 | BAU estimation in INDCs (not ratified yet) | 928 |
| USA | Reduce GHG emissions by 26–28% below its 2005 level in 2025 | assuming linear rate [50] | 4200–4400 |
| Canada | Reduce GHG emissions by 30% below 2005 levels by 2030 | 2005 emissions, UNFCCC database | 512 |
| Mexico | Reduce GHG emissions by 22–36% below BAU of 973 Mt. | BAU is given in NDCs | 623–759 |
| Australia | Economy-wide target to reduce GHG emissions by 26–28% below 2005 levels by 2030 | 2005 emissions, UNFCCC database | 402–414 |
| Japan | reduction of 26% by fiscal year (FY) 2030 compared to FY 2013 | 2013 emissions from NDCs | 771 |
| New Zealand | reduce GHG emissions to 30% below 2005 levels by 2030 | 2005 emissions, UNFCCC database | 57.8 |

**Table 6.** *Cont.*

| Region or Country | Summary of NDCs | Methodology | Mt CO$_2$e in 2030 |
|---|---|---|---|
| Republic of Korea | Emission reduction by 37% from BAU level by 2030 | BAU estimation given in NDCs | 536 |
| China | Peak emissions before 2030 + Lower CO$_2$ emissions per unit of GDP by 60–65% from 2005 level | assuming SSP2 [50] | 15900–17600 |
| India | Reduction in emissions intensity of GDP by 33–35% from 2005 levels by 2030 | assuming SSP2 [50] | 6900–7100 |
| Russia | Limiting anthropogenic GHG to 70–75% of 1990 levels by 2030 | 1990 emissions, UNFCCC database (still INDCs) | 1757–1887 |

* Indicates Intended NDCs.

The values for GHG emissions in 2030 implied by these NDCs are shown in Figure 7 for six different regions (OECD, OECD Europe, OECD North America, China, India, and Russia), assuming the ranges shown on Table 6, and were compared with the CO$_2$ emissions from the energy sector estimated in the WEO reports of 2006, 2010, 2015, 2017, and 2018. The time interval between the first and last of these five WEO reports is equal to that between 2018 and 2030. We are perfectly aware of comparing different things in this exercise. More specifically, the WEO projections considered here do not account for other non-CO$_2$ GHG gases, which would certainly inflate total GHG emissions, and LULUCF is also not accounted. Still, we see that for most regions the NDCs values are comparable with the IEA "Reference" (2006) and "Current Policies" (2010, 2015, 2017, 2018) scenarios. As an example, for the whole of the OECD region, the WEO in 2006 and 2010 was still projecting CO$_2$ emission levels for the energy sector which were superior to the total value of the GHG emissions (average between minimum and maximum range) these regions committed to for 2030 (Figure 7). Only in the case of China and India were the NDCs estimations considerably higher than the CO$_2$ emissions from the energy sector only projected by the WEO reports between 2006 and 2018.

For comparison, we also considered the maximum variation ("maxVar") found for the CO$_2$ emissions from the energy sector in 2030, between the considered set of (seven) 2006–2018 WEO reports (Figure 4, middle row). Figure 7 shows the result of adding and subtracting these variations to the IEA WEO 2018 projections (the most recent ones) for 2030. We found that the CO$_2$ emission projections (without any NDCs commitments made by countries, which in the WEO reports are part of the "New Policies" Scenario, and thus are not considered here) are thus well within the range of the NDCs, except in the case of China. The reason for this can be explained by the fact that since 2006 the WEO projections for CO$_2$ emissions have varied very little. For India, adding this maximum variation to the WEO 2017 values puts it very close to the average NDCs estimation. It should be noted that these two regions have very high levels of uncertainty regarding future developments [29,50]. In the case of Russia, its INDCs commitment is by all accounts fairly unambitious, due to the very high level of their GHG baseline emissions in 1990, and so it is hard to conclude much from this comparison.

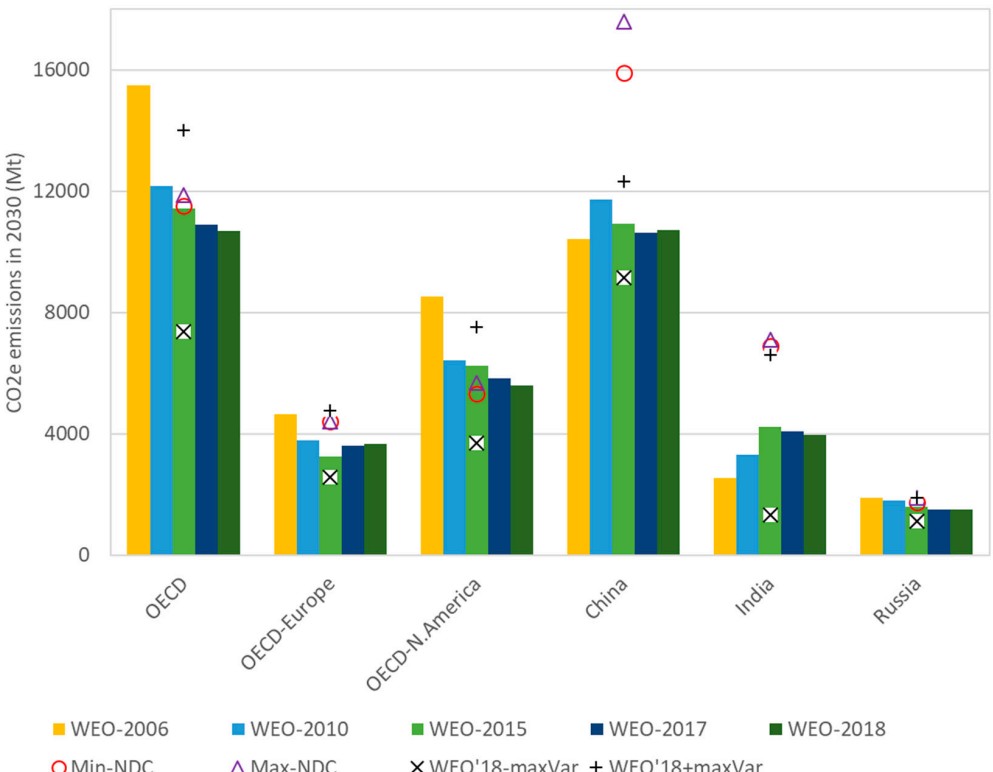

**Figure 7.** WEO projections (2006, 2010, 2015, 2017, and 2018) for $CO_2$ emissions for the energy sector in 2030 and the NDC ranges (minimum and maximum) for the proposed GHG emissions in 2030, for 6 different regions. The label "maxVar" indicates the magnitude of the maximum variation found between the projections for these regions (Figure 4b). These are 31%, 30%, 35%, 15%, 66%, and 25%, respectively. The variations were then subtracted and added to the respective WEO 2018 $CO_2$ emissions projection for 2030, for comparison.

Thus, we find that the answer to the initial question is clearly affirmative, i.e., the magnitude of the variations found in the WEO projections for $CO_2$ (the main GHG) emissions from the energy sector for the six different regions shown in Figure 7 is indeed higher or equal to the global gap of 27% between current NDCs and the 2 °C Paris Agreement target for 2030, except in the case of China. However, we should also note the different direction in which these variations occurred, which was negative for the case of OECD, OECD Europe, OECD North America, and Russia (implying a downward correction in emission projections), but positive in the case of China and India, implying an upwards correction (Figure 4b).

## 4. Discussion

Our analysis of TPED, $CO_2$ emissions from the energy sector and the percentage of the RES electricity projections in thirteen IEA World Energy Outlook reports published between 1994 and 2018 has revealed some underlying trends. In particular, towards slightly overestimating TPED (and subsequent $CO_2$ emissions) for some regions, as well as systematically underestimating the rate of implementation of renewable energy sources. These are briefly summarized in Table 7.

**Table 7.** Main summary of the results for the 8 regions and the 3 main variables analyzed in Section 3.2, as regards: (i) difference between projections and observed values (small variations, significant variations; underestimated, overestimated); (ii) future trends for 2040 (reduction, constant, small growth, large growth); and (iii) recent changes to projected trends for 2030 and 2040 (revised downwards, revised upwards).

| Region | TPED | CO$_2$ Emissions from the Energy Sector | Percentage of RES Electricity |
|---|---|---|---|
| World | • small variations<br>• growth projected<br>• revised downwards | • significant variations<br>• large growth projected<br>• revised downwards | • underestimated<br>• growth projected<br>• revised upwards |
| OECD | • overestimated<br>• projected constant<br>• revised downwards | • overestimated<br>• reduction projected<br>• revised downwards | • underestimated<br>• growth projected<br>• revised upwards |
| OECD Europe | • small variations<br>• projected constant<br>• revised upwards | • small variations<br>• reduction projected<br>• revised upwards | • underestimated<br>• growth projected<br>• revised upwards |
| OECD N. America | • overestimated<br>• small growth projected<br>• revised downwards | • overestimated<br>• reduction projected<br>• revised downwards | • underestimated<br>• growth projected<br>• revised upwards |
| China | • underestimated<br>• growth projected<br>• revised downwards | • underestimated<br>• growth projected<br>• revised downwards | • Significant variations<br>• small growth projected<br>• revised upwards |
| India | • significant variations<br>• large growth projected<br>• revised downwards | • significant variations<br>• high growth projected<br>• revised downwards | • overestimated<br>• high growth projected<br>• revised upwards |
| Russia | • small variations<br>• small growth projected<br>• revised upwards; | • overestimated<br>• small growth projected<br>• revised downwards | • underestimated<br>• small growth projected<br>• revised downwards |
| Africa | • underestimated<br>• growth projected<br>• revised upwards | • small variations<br>• high growth projected<br>• revised upwards | • underestimated<br>• high growth projected<br>• revised upwards |

As discussed in previous work (e.g., the authors in [4,5,9,52,53]), one possible explanation for this discrepancy is that the legacy models for forecasting energy demand and new technology implementation are likely ill-suited for dealing with disruptive events or very fast transitions, including the kind of accelerated transition that is now required worldwide in order to drastically reduce GHG emissions. This mismatch might be thrown into sharper focus in a situation where demand for global climate action spreads worldwide and overcomes most short-term economic considerations, forcing governments and companies into an accelerated path of emissions reduction. One possible way to improve energy systems modeling is by promoting a wider adoption of agent-based modeling of energy systems and energy transitions [54–56], in combination with insights from behavioral science and a mature view of real-life stakeholder decision processes [53].

In terms of other methodological aspects which could be driving these results, we believe one major issue is very likely: the general assumption of persistent high GDP growth rates, both in global terms, as well as for specific regions of the world, in the foreseeable future, as discussed by previous authors, thus also driving a high growth of TPED [24]. As an example, in the 2018 description of the underlying simulation model for the WEO scenarios, we can read that "In WEO-2018, world gross domestic product (GDP) is expected to grow on average by 3.4% per year over the projection period (2017–2040) That rate is slower than past trends (3.6% in 2000–2017)" [44]. It could be argued that such growth rates might be very hard to maintain in that 23-year period, particularly at a time of (expected) increasing commodity shortages, higher environmental impacts, and more extreme weather events. Moreover, the regions of the world whose economic growth will be leading world GDP growth in the period 2017–2040 are also in many cases the same

regions which will feel the worse impact of climate change, e.g., Africa, the Middle East, and Asia Pacific. For these regions, the projected GDP growth rates are 4.3%, 3.4%, and 4.5%, respectively, for that period in the central WEO scenario of 2018 [44].

Several NDCs (e.g., for Mexico, Republic of Korea, Turkey) are made in relation to so-called business-as-usual (BAU) scenarios, where typically very high economic growth is assumed, with a corresponding increase in energy demand and emissions. Oftentimes, these BAU scenarios are based on assumptions stemming from relevant and recognized sources for energy policy and analysis, which include the IEA's WEO. As mentioned by other authors, we argue that this can in turn create a kind of "self-fulfilling prophecy" [9,11], which can work either in the direction of more sustainable energy systems or in an opposite direction [53].

Another example of possible methodological biases are the assumptions behind fossil fuel subsidies, which in the World Energy Model (WEM) documentation [44] are also expected to continue to exist in the foreseeable future, something which is widely seen as incompatible with the Paris Agreement goals [25]. Carbon prices are also relevant to thisdiscussion, being expected to be USD 22 per tonne in the IEA Current Policies Scenario and USD 25 in the New Policies Scenario by 2025, for the European Union [44]. However, in the last three quarters of 2019, the EU Emissions Trading Scheme (ETS) carbon price never fell below EUR 21 per tonne, and thus was already consistently trading higher for most of the time than the price considered in the WEM only for 2025. More incredibly, the EUR 50 per tonne threshold was breached in May 2021 (URL: https://ember-climate.org/data/carbon-price-viewer/ (accessed on 31 May 2021)). This could in turn affect the calculations in terms of which power plants are still profitable, in particular coal and natural gas ones, drastically reducing their expected lifetime [24,57].

## 5. Conclusions

Whilst the IEA projections are seen to be fairly consistent on a global scale (particularly for TPED and $CO_2$ emissions from the energy sector), this consistency can actually mask very large regional variations. Thus, we found the maximum variation in $CO_2$ projections (from the energy sector) from the period 2006–2018 to be comparable to the additional GHG emissions reductions needed to close the gap between current NDCs and the 2030 emission targets. More concretely, the current (global) gap between existing NDCs and the reduction in GHG emissions needed to comply with the Paris Agreement goal of warming below 2 °C was recently estimated at 15 $GtCO_2$e, or 27% of 2030 GHG emissions [30]. In this analysis we found variations of −31%, −30%, and −34%, for the OECD, OECD Europe, and OECD North America regions, respectively, in $CO_2$ emissions from the energy sector projected for 2030 by different WEOs published in the last twelve years (Figure 4b). For China, India, and Africa these values were +15%, +66%, and −28%, respectively, for the same time interval, albeit in a different direction for the first two countries.

Thus, when we directly compared the estimated NDCs with the values of $CO_2$ emissions (from the energy sector) projected by the IEA for 2030 (Figure 7), we found that for four of the regions analyzed (OECD, OECD Europe, OECD North America, and Russia), the values were very similar. When taking into account the maximum variation observed in projections made over a period of 12 years, then even in the case of India, the NDCs commitments fell within the range of recent WEO variations or corrections, albeit at the higher end of the spectrum. This in turn raises the question that the IEA projections for $CO_2$ emissions from the energy sector, at least in the "Current Policies" scenario, might be considerably inflated [12].

This is a striking and unexpected finding, which bears repeating, as we find that even within the most conservative scenarios (not including commitments from NDCs), WEO projections in a period of 12 years can still have such a relative variation. It may also indicate that the general transformation in global energy systems required to meet the 2 °C target may be slightly less difficult than has been suggested.

We note, however, that in this work the more sustainable "450" or "Sustainable Development" and the "New Policies" scenarios (the latter already taking into account NDCs policy), were not considered. One thing that this analysis seems to suggest is that perhaps the so-called Reference (or BAU) scenarios should already include not only all adopted policies, but the policies strongly expected to be implemented in the short to medium term as well [58]. By turning the "New Policies" scenario into the "Reference" or baseline, this could have an important psychological effect on policy makers and stakeholders around the world. In this context we note that although the "Current Policies" should obviously be read as: "if nothing else changes, and under such and such assumptions, this is what we can expect to happen" they are much more often utilized by several large companies and governments around the world as "this is what is most probable to happen". In this respect, these projections have been used as an argument for all types of (extreme) fossil fuel expansion projects, including coal mining in Australia, Alaskan offshore oil drilling, and Canadian Tar Sands, under the guise of satisfying projected energy demand [12].

Furthermore, the differences between regions, such as the OCED and India and Africa, should perhaps be acknowledged more carefully in the modeling exercise itself. As an example, in the case of OECD Europe many of the main issues revolve around restructuring an energy system already established many decades ago and deciding what the best financial incentives are for decommissioning the most polluting power plants and industrial facilities [59]. In contrast to this, in regions such as Africa, India, and parts of Southeast Asia, many of the relevant investment decisions regarding energy systems have not yet been made, and issues such as air pollution, high child mortality rates, or stranded assets could play a much more relevant role, thus tilting the balance towards renewable energy generation [17,60].

**Author Contributions:** L.M.F. (conceptualization, methodology, data analysis, original draft preparation) and S.G.S. (conceptualization, software, validation, writing—review and editing, supervision). All authors have read and agreed to the published version of the manuscript.

**Funding:** This research was funded in part by the Portuguese Science and Technology Foundation (FCT), under research grant PD/BD/128171/2016. Also, the research work developed at CENSE is financed by FCT through the strategic project UIDB/04085/2020.

**Institutional Review Board Statement:** Not applicable.

**Informed Consent Statement:** Not applicable.

**Data Availability Statement:** Not applicable.

**Acknowledgments:** We acknowledge comments from the participants of the 73rd ETSAP (Energy Technology Systems Analysis Program) Semi-Annual Meeting, which took place at Chalmers University of Technology in Gothenburg, Sweden, on June 2018, as well as helpful criticism from Júlia Seixas (NOVA University Lisbon), colleagues from CENSE and Maria Fernandes-Jesus (ISCTE-IUL). We are also grateful to the three anonymous reviewers, whose comments and recommendations greatly improved and helped clarify the original manuscript.

**Conflicts of Interest:** The authors declare no conflict of interest.

## Appendix A

Table A1 shows the regions/countries considered in this work (besides world: OECD, OECD Europe, OECD North America, China, India, Russia, and Africa) and the corresponding countries in the case of regional aggregations. Large GHG emitters, such as Brazil and Indonesia, were left out of this analysis due to the very large impact of land use, land-use change, and forestry (LULUCF) in their GHG emission trends, and thus the smaller relative importance of energy-related GHG emissions 30]. The definition of regional groupings of countries is strictly the one adopted by the IEA in each version of the WEO. This is because we were comparing the results of the IEA's energy and $CO_2$ emission projections across time and with historical values. To do so, we were limited by the often-changing definition of regions considered in the WEO. OECD Pacific countries

are shown since these are also accounted for in the sum of OECD countries. Missing from the list of OECD countries is Chile since the South America region was not separately considered in this work.

Starting from 2017, there was a significant change in the WEO regional definitions, and the OECD Europe group was no longer considered [42]. Instead, there was a new regional definition called "Europe", consisting of all the (by then) 28 countries of the EU (of these, Bulgaria, Croatia, Cyprus, Latvia, Lithuania, Malta, and Romania were not previously included in the "OECD Europe" grouping) plus Albania, Belarus, Bosnia and Herzegovina, Gibraltar, Kosovo, Montenegro, Serbia, the Former Yugoslav Republic of Macedonia, the Republic of Moldova, and Ukraine as well as Iceland, Israel, Norway, Switzerland, and Turkey, which were previously also included in the "OECD Europe" regional definition.

As for the other four regions not included in the several OECD definitions, the data for Russia were only considered starting from the WEO 2000, since before that it was grouped under the heading of either "Countries of the former Soviet Union" (WEO 1994, 1996) [31,32] or "Transition Economies" (WEO 1998) [28]. We note that in a group consisting of 15 countries in total, the Russian Federation emissions accounted for 63%, 66%, and 66% of $CO_2$ emissions, in 1994, 1996, and 1998, respectively, according to data from www.iea.org/statistics (accessed on 25 October 2020). Other large emitters during this period include Ukraine, Kazakhstan, and Uzbekistan.

In the WEO reports of 1994, 1996, and 1998 India appears grouped under the "South Asia" definition [31–33], and we used those values as a proxy for India, due to its very large size compared to the other countries in that group (Pakistan, Bangladesh, Sri Lanka, and Nepal). Starting from the WEO in 2000, the values for India were shown separately in all the WEOs and were thus considered in the present analysis [34]. In this group of five countries, and for the years 1994, 1996, and 1998, India's $CO_2$ emissions accounted for approximately 87% of the total, its TPED for about 82%, its share of population was 77%, and the share in electricity consumption varied between 85% and 86% (www.iea.org/statistics and World Bank data, accessed on 25 October 2020).

In the case of China, we strictly adhered, once again, to IEA definitions. Exact definitions can be found at the end of each WEO (Section "Regional definitions"), and it is routinely considered that data for China includes Hong Kong as well (since 1997), but that "Chinese Taipei" is not included. Finally, the list of countries under the regional definition of "Africa" has remained constant throughout the years.

**Table A1.** Variation of composition of regional definitions considered within the analyzed WEO reports.

| WEO Edition | OECD North America | OECD Europe/Europe (after 2017) | OECD Pacific | Africa | China | South Asia/ India | Russia |
|---|---|---|---|---|---|---|---|
| | | | **Regional Definitions** | | | | |
| 1994 | Canada and United States | Austria, Belgium, Denmark, Finland, France, Germany, Greece, Iceland *, Ireland, Italy, Luxembourg, Netherlands, Norway, Portugal, Spain, Sweden, Switzerland, Turkey and United Kingdom | Australia, Japan and New Zealand | | | South Asia (including Pakistan, Bangladesh, Sri Lanka and Nepal) | Former Soviet Union (not considered in this analysis) |
| 1996 | | | | | | | |
| 1998 | | + Czech Republic and Hungary | | | | | |
| 2000 | | | | | | | |
| 2002 | | +Poland | | | | | |
| 2004 | | | | | | | |
| 2006 | | +Slovak Republic | | | | | |
| 2008 | | | + Korea | All African countries | China | India | Russia |
| 2010 | +Mexico | | | | | | |
| 2012 | | + Estonia and Slovenia * | | | | | |
| 2015 | | +Israel * | | | | | |
| 2017 | | + Bulgaria *, Croatia *, Cyprus *, Latvia *, Lithuania *, Malta *, Romania *, Albania *, Belarus *, Bosnia and Herzegovina *, Gibraltar *, Kosovo *, Montenegro *, Serbia *, the Former Yugoslav Republic of Macedonia *, the Republic of Moldova * and Ukraine * | | | | | |
| 2018 | | | | | | | |

Note: OECD country Chile is not included (see text). * Indicates non-IEA membership as of 2019. + Denotes that the countries it precedes were added to the group of countries in the respective region.

## Appendix B

In order to better analyze the trends found in this work, we took into consideration the general evolution of several economic indicators in the past decades for the eight regions considered. Table A2 thus shows the values for: population, $CO_2$ emissions from the energy sector, and GDP (in USD in 2010, PPP) for the eight regions, as well as their evolution between 1990 and 2013.

**Table A2.** Overview of significant indices for the analyzed regions, between 1990 and 2013.

| Region | Population (Millions) | | % Variation '90–'13 | CO₂ Emissions from Energy (Mt CO₂e) | | % Variation '90–'13 | GDP PPP (USD Billion in 2010) | | % Variation '90–'13 |
|---|---|---|---|---|---|---|---|---|---|
| | 1990 | 2013 | | 1990 | 2013 | | 1990 | 2013 | |
| World | 5288 | 7185 | 36% | 20.518 | 32.288 | 57% | 46.097 | 98.997 | 115% |
| OECD | 1069 | 1265 | 18% | 11.020 | 12.031 | 9% | 29.088 | 45.719 | 57% |
| OECD Europe | 543 | 596 | 10% | 3924 | 3563 | −9% | 12.307 | 18.697 | 52% |
| OECD N. America | 363 | 474 | 31% | 5508 | 6187 | 12% | 11.047 | 19.524 | 77% |
| China | 1135 | 1357 | 20% | 2089 | 9191 | 340% | 1698 | 15.894 | 836% |
| India | 870 | 1279 | 47% | 530 | 1853 | 249% | 1496 | 6356 | 325% |
| Russia | 148 | 144 | −3% | 2163 | 1514 | −30% | 2715 | 3201 | 18% |
| Africa | 629 | 1122 | 78% | 529 | 1086 | 105% | 2106 | 5012 | 138% |

Note: Population data from World Bank Open Data: (1) United Nations Population Division. World Population Prospects: 2017 Revision. (2) Census reports and other statistical publications from national statistical offices, (3) Eurostat: Demographic Statistics, (4) United Nations Statistical Division. Population and Vital Statistics Report (various years), (5) US Census Bureau: International Database, and (6) Secretariat of the Pacific Community: Statistics and Demography Program. Available at: https://data.worldbank.org/indicator/SP.POP.TOTL (accessed on 25 October 2020); Energy and emission data from IEA Energy balances and indicators. Available at: https://www.iea.org/statistics (accessed on 25 October 2020).

As seen in Table A2, the different regions vary greatly and had very different trajectories. Africa, for example, had the largest population growth between 1990 and 2013 (left columns), double that of the relative increase in the world in the same period (78% versus 36%, respectively). However, its GDP only grew slightly above the world average (right columns), 138% versus 115%, respectively. China, India, and Africa experienced the largest growth in $CO_2$ emissions from the energy sector between 1990 and 2013 (middle columns, 340%, 249%, and 105%, respectively), whereas emissions decreased slightly in OECD Europe during the same period (−9%) and by a substantial factor for Russia (−30%). In the same period, China's GDP grew by a factor of 9.36 and India's by a factor of 4.24 (right columns).

Table A3 then combines the indicators shown in Table A2 with energy statistics for the same time period, looking at the variation of TPED, electricity consumption (both in units of MWh), and $CO_2$ emissions from the energy sector, in terms of per capita (top half) and intensity per USD one thousand in 2010 (bottom half). These three specific fields are closely related to the ones analyzed in the rest of this work.

From Table A3 we see that nearly all indicators improved their performance over that time period in terms of economic efficiency (i.e., decreased intensity per unit of economic value). The only exception occurred for electricity intensity in China, with an increase of 6%. In terms of per capita indicators, the situation is more mixed. For example, the very modest reductions made in TPED per capita in the OECD, OECD Europe, and OECD North America (−1%, −3%, and −9%, respectively) can be contrasted with the huge increases in China and India (179% and 73%, respectively), and the same can be said in terms of $CO_2$ emissions per capita, with OECD Europe registering the largest decrease (−17%). In this period, all regions increased their electricity consumption per capita, with the exception of Russia (−7%). In the case of Africa, its per capita increases were all below the world average, which can be in part explained by the very large population growth (Table A2) not being accompanied by a proportional growth in energy usage [17,45].

**Table A3.** Key energy indicators for the studied regions, based on IEA Energy Statistics.

| Region | TPED per Capita (MWh/Capita) | | % Variation '90–'13 | Electricity per Capita (MWh/Capita) | | % Variation '90–'13 | $CO_2$ Emissions from Energy per Capita (t $CO_2$e/Capita) | | % Variation '90–'13 |
|---|---|---|---|---|---|---|---|---|---|
| | **1990** | **2013** | | **1990** | **2013** | | **1990** | **2013** | |
| World | 19.30 | 22.27 | 15% | 1.83 | 2.72 | 48% | 3.88 | 4.49 | 16% |
| OECD | 49.31 | 48.85 | −1% | 6.01 | 7.44 | 24% | 10.31 | 9.51 | −8% |
| OECD Europe | 34.85 | 33.97 | −3% | 4.12 | 5.14 | 25% | 7.22 | 5.98 | −17% |
| OECD N. America | 72.58 | 65.93 | −9% | 8.73 | 9.69 | 11% | 15.18 | 13.05 | −14% |
| China | 8.95 | 24.94 | 179% | 0.40 | 3.31 | 729% | 1.84 | 6.77 | 268% |
| India | 4.09 | 7.08 | 73% | 0.25 | 0.68 | 176% | 0.61 | 1.45 | 138% |
| Russia | 68.96 | 58.34 | −15% | 5.57 | 5.19 | −7% | 14.59 | 10.55 | −28% |
| Africa | 7.24 | 7.86 | 9% | 0.41 | 0.53 | 30% | 0.84 | 0.97 | 15% |

| Region | TPED Intensity (MWh/1000 USD 2010) | | % Variation '90–'13 | Electricity Intensity (MWh/1000 USD 2010) | | % Variation '90–'13 | Carbon Intensity (t $CO_2$e/1000 USD 2010) | | % Variation '90–'13 |
|---|---|---|---|---|---|---|---|---|---|
| | **1990** | **2013** | | **2013** | **1990** | | **1990** | **2013** | |
| World | 2.21 | 1.62 | −27% | 0.21 | 0.20 | −6% | 0.45 | 0.33 | −27% |
| OECD | 1.81 | 1.35 | −25% | 0.22 | 0.21 | −7% | 0.38 | 0.26 | −31% |
| OECD Europe | 1.54 | 1.08 | −30% | 0.18 | 0.16 | −10% | 0.32 | 0.19 | −40% |
| OECD N. America | 2.38 | 1.60 | −33% | 0.29 | 0.24 | −18% | 0.50 | 0.32 | −36% |
| China | 5.98 | 2.13 | −64% | 0.27 | 0.28 | 6% | 1.23 | 0.58 | −53% |
| India | 2.38 | 1.42 | −40% | 0.14 | 0.14 | −4% | 0.35 | 0.29 | −18% |
| Russia | 3.77 | 2.62 | −31% | 0.30 | 0.23 | −24% | 0.80 | 0.47 | −41% |
| Africa | 2.16 | 1.76 | −19% | 0.12 | 0.12 | −3% | 0.25 | 0.22 | −14% |

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
