# Peer review of "Historical Variation of IEA Energy and CO2 Emission Projections: Implications for Future Energy Modeling"

_sustainability, doi:10.3390/su13137432_

Round 1
Reviewer 1 Report
The paper is interesting and well written. I suggest it can be improved and made more readable by decreasing the number of numbers that are quoted in the text and in the tables. For instance, in tables 2,3,4, I’d prefer to see only the average value of the projections compared to historical variations. The max and min values are not clearly defined and belong to reports issued at different times, so it is a bit hard to interpret them correctly. Additionally, they are all related to the same intermediate scenario while one may think they represent different alternatives (as in the IPCC paths). Additionally, using max and min with negative numbers, as in table 5, may be misleading, since it is not clear if we refer to the max and min values or to the max and min variations. For instance, in table 5, a min value of -43% is indicated as closest to the real variation of -35% (for CO2 emission), whereas the closest appears to be the average of -30%. In the same way, I would have marked as closest to the Chinese total ELC of -1% the average value of -9% instead of the maximum of 7%. While the distance appear to be the same, at least the average has the correct sign.
On the same line, I’d prefer to see just a range of uncertainty (a band including all the forecasts) in figures 1,3,5 compared to the historical development. As they are now, they are very difficult to read and it is almost impossible to catch all the information.
Remaining on the figures, I think it is better to see figure 2,4,6 all with the same scale. Otherwise, the reader may be confused, despite this difference is pointed out in the caption.
As a general comment, I think that table 4 and 5 are the core of the problem. The energy consumption is in fact basically related to the population and the intensity. But population changes slowly in absolute values and age composition, which means that population models can be very precise 10 or 20 year ahead. On the contrary, as noted by the authors, the energy intensity and the energy mix are changing much more rapidly because technology evolves at a difference pace wrt population. In particular, what is behind table 5, and not sufficiently underlined, is the world movement toward the reduction of energy consumption in all sectors from buildings to industry. This means (assuming the GDP as a measure of richness despite its well-known limitations) enjoying the same services with less energy. I suggest the authors reduce a bit all the numbers and concentrate more on these conceptual aspects.
Minor comments:
Line 102 “potential energy demand destruction might be systematically underestimated“ what do you mean by “destruction”?
Line 187 PPP appears here, but is defined later at 331
Line 187 “maximum, average and minimum variations” here they are defined in terms of variations, but then they seem to be in term of values, when the variation is negative.
Line 259 TPED was already defined
Line 334 “if were to normalize the variations found in the TPED and CO2 334 projections” rewrite the sentence
Line 408 repeats what state at line 401
Line 428-432 this and similar very long sentences should be avoided.
Reviewer 2 Report
The study carries out thirteen sets of World Energy Outlook projections to assess how well its projections are aligned with sustainable development goals as well as closely tracking observed, historical values from 1994 to 2018.
First, the topic is interesting and the methodology is suitable for the readers. The authors offer plentiful data and logical explanations.
Second, I have some suggestions for this paper to be clear for readers as follows, in the Conclusions section, I suggest that it would be clear to have a table with different groups from scenario's projections analysis in this article.
Reviewer 3 Report
Overall impression:
The manuscript is certainly interesting based on the proposal to analyze the World Energy Outlook reports, produced by the International Energy Agency, I am very familiar with, because it is in fact considered as a “gold standard” for researchers and decision takers. I can say that I really enjoyed the reading and reviewing process.
In mi opinion the manuscript looks almost finished, properly written, a bit large but concise. Minor revisions remain before the manuscript can be accepted. I would like to say that the present manuscript is highly interesting and contributes to the scientific research domain, congratulations by this initiative.
I hope my suggestions below help the authors to add some points to improve the whole manuscript.
Minor issues.
- I notice an issue with the “balance” of manuscript, results start in page 4, and finish in page 19 (full pages from page 5 to 18, 14 pages), compared to just 26 lines of discussion to conclude with (an unusual, extended section) almost 2 pages. So, try to improve the discussion section and reduce the conclusions sections, my guess is the results section is long enough, interesting and herd to reduce.
